


**Analytical solution of the ray equations of Hamilton for Rossby waves on stationary shear**

2                                              **flows**

Vladimir G. Gnevyshev[1] and Tatyana V. Belonenko[2] *
[1]Shirshov Institute of Oceanology Russian Academy of Science, 117997, Nakhimovskiy pr., 36, Moscow, Russia

       [2]St. Petersburg State University, 199034, Universitetskaya emb., 7-9, St. Petersburg, Russia

Correspondence to and Tatyana V. Belonenko (btvlisab@yandex.ru)
**Abstract**
The asymptotic behavior of Rossby waves in the ocean interacting with a shear stationary flow is
considered. It is shown that there is a qualitative difference between the problems for the zonal
and non-zonal background flow. Whereas only one critical layer arises for a zonal flow, then
several critical layers can exist for a non-zonal flow. It is established that the integrated ray
equations of Hamilton are equivalent to the asymptotic behavior of the Cauchy problem solution.
Explicit analytical solutions are obtained for the tracks of Rossby waves as a function of time and
initial parameters of the wave disturbance, as well as the magnitude of the shear and angle of
inclination of the flow to the zonal direction. On the example of Rossby waves on a shear flow,
the ray equations of Hamilton are analytically integrated. The obtained explicit expressions make
it possible to calculate in real-time the Rossby wave tracks for any initial wave direction and any
shear current inclination angle. It is shown qualitatively that these tracks for a non-zonal flow are
strongly anisotropic.
**Keywords:**
Rossby waves, shear flow, zonal, non-zonal, Hermitian operators, Non-Hermitian operators, ray
equations of Hamilton

24         **1.  Introduction**

Historically, the problem of studying the interaction of Rossby waves with large-scale

currents began with problems for the atmosphere, in a formulation in which the large-scale
background flow was considered strictly zonal (Rossby et al., 1939). This formulation is quite
justified for the atmosphere. Rapid advances in satellite altimetry have contributed to the rapid
development of empirical understanding of Rossby waves in the ocean (Fu and Cazenave, 2000).
Analysis of the variability of oceanological fields confirms the existence of Rossby waves in the
World Ocean. However, unlike the atmosphere, Rossby waves in the ocean have their specifics.



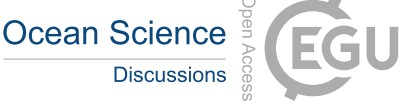

The main difference is that in the ocean, background currents are usually not zonal. Moreover, the
strongest dynamic processes occur on non-zonal flows or when the initial zonal flow deviates from
the zonal direction as observations show (Gnevyshev et al., 2020a, b).

One of the central moments in the interaction of Rossby waves and large-scale flows are

critical layers. The classical critical layer is not formally attainable for waves. It is the geometric
border of the transparency region and the shadow region. The critical layer is defined as $c = U$, i.e.
the equality of the longitudinal component of the phase velocity of the wave c and the velocity of
the background current $U$. The critical layers have been studied and are well known for
gravitational waves and internal waves (LeBlond, Mysak, 1978). For Rossby waves, the study of
the critical layer historically also began with the zonal critical layer.

If the background current is strictly zonal, then, as shown in (Gnevyshev et al., 2020a), the

determination of the critical layer through the phase velocity is quite correct and can be applied
for Rossby waves. However, if the flow is not zonal, such a definition becomes ambiguous and
allows Rossby waves to cross the critical layer, with the formation of the so-called overshooting
effect. The propagation of Rossby waves on shear flows has its specific feature: the wave track
gradually approaches its critical layer, this occurs asymptotically for a long time.

One of the features of Rossby waves is the qualitative difference between the problems for

the zonal and non-zonal background flow. The first key point that distinguishes the problems of a
zonal background flow and a critical layer from a non-zonal one is the number of critical layers.
For a strictly zonal flow, there is only one critical layer, while for a non-zonal shear flow, three
qualitatively different cases can be distinguished (Gnevyshev et al., 2020a, b) we will consider a
bit later. As a consequence, the passage to the limit from a non-zonal flow to a strictly zonal case
is nontrivial. In particular, all asymptotic laws under the passage to the limit are of a discontinuous
nature (Gnevyshev et al., 2020a, b). In this case, of the three non-zonal critical layers in the passage
to the limit, from the non-zonal to the zonal critical layer, only one critical layer remains. And the
transition from the zonal to the non-zonal case, in principle, is not possible. As a consequence, a
strictly meridional flow acquires the most general character, rather than a purely zonal flow.

The second important point for Rossby waves is that the linear operator of Rossby waves

ceases to be Hermitian upon passing to the non-zonal case. The adiabatic invariant in the form of
the enstrophy conservation law, which exists in the WKB approximation, ceases to hold for non-
zonal piecewise linear flow profiles of the "vortex layer" type. A non-zonal strong shear current
enters into an active exchange of vorticity with Rossby waves (LeBlond and Mysak, 1978;
Fabrikant, 1987; Stepanyants and Fabrikant, 1989; Gnevyshev and Shrira, 1990).




The fundamental point in which the analysis of problems for the ocean differs from the
atmosphere is the limitedness of ocean currents in space and, as a consequence, in time. Therefore,
for the obtained qualitative results of the analysis of the dynamics of Rossby waves to have an
applied character, it is important to understand what periods and spatial scales are behind such
concepts as "approaching" the critical layer?
The classical approach for analyzing the kinematics of waves in dispersive systems is based
on the ray equations of Hamilton. However, as is customary even in classical mechanics, no one
explicitly solves the differential equations of Hamilton in analytical form. The traditional approach
is qualitative and is based on the presence of cyclic variables in the problem. As a rule, these are
the longitudinal component of the impulse and the frequency of the wave. If we also use a certain
set of symmetries, related to the Hermitian nature of the linear wave operator, then this purely
geometric approach suffices to understand qualitatively the evolution of waves on plane-parallel
inhomogeneous flows, without solving the ray equations of Hamilton explicitly. Therefore, it is
better to use a qualitative method, which is called the isofrequency method. It is based on the
geometric construction of isofrequency lines and the concept of the direction of the group velocity.
For Rossby waves, a qualitative analysis of the kinematics based on the isofrequency method was
performed as early as (Ahmed, Eltaeb, 1980; Duba et al., 2014).

Based on the fact that asymptotically long adhesion of Rossby waves to the critical layer
has already been established, we are trying to understand the specific features of this process. The
goal of our work is to determine how real the periods and spatial scales of this process are so that
they can be realized for real conditions in the ocean. To answer this question, it is necessary to
have explicit analytical solutions for wave tracks as a function of time and initial parameters of the
wave disturbance, as well as the magnitude of the shear and the angle of inclination of the flow to
the zonal direction. In addition, in this paper, using the example of Rossby waves on a shear flow,
we analytically integrate ray equations of Hamilton for the first time. The obtained explicit
expressions make it possible to calculate in real-time the Rossby wave tracks for any initial wave
direction and any shear current inclination angle. As will be shown below, such tracks for a non-
zonal flow are qualitatively highly anisotropic.
The generally accepted way to obtain a solution as a function of the initial position of the
wave and time is to solve the Cauchy problem. For barotropic Rossby waves, the Cauchy problem
was solved in (Yamagata, 1976a, b) for strictly zonal and meridional currents. Continuing this
direction, we will show that the integrated ray equations of Hamilton turn out to be equivalent to



98 the asymptotics of the solution of the Cauchy problem. However, in contrast to (Yamagata, 1976a,

99 b), we propose an easier way to obtain explicit analytical expressions for the Rossby wave tracks.

100 To obtain a solution, the introduction of convective coordinates, direct and inverse Fourier

101 transforms, and the stationary phase method for the obtained two-dimensional Fourier integral is

102 not required (Yamagata, 1976a, b). In this work, we will show that ray equations of Hamilton for

103 Rossby waves are integrated with explicit expressions quite simply using the arctangent and

104 logarithm functions, in contrast to the solutions of Yamagata (1976a, b), which use a more specific

105 mathematical apparatus related to the Cauchy problem. The new solutions of the ray equations of

106 Hamilton for Rossby waves are much simpler than the geometric method of isofrequencies and

107 represent explicit analytical expressions for the tracks of Rossby waves in elementary functions.

### 2. Results

110   The ray equations of Hamilton are an effective tool for analyzing the kinematic properties

111 of Rossby waves in a plane-parallel shear flow (LeBlond, Mysak, 1978; Salmon, 1998). In

112 practice, this method is often successfully applied in numerical calculations (see, for example,

113 Killworth & Blundell, 2003). We will show that for shear flows there is also an explicit analytical

114 solution of these equations, and these solutions will be found in elementary functions. The so-

115 called equations of geometric optics are as follows:

$$116 \qquad k_t = -\frac{\partial \omega}{\partial x}, \quad l_t = -\frac{\partial \omega}{\partial y}, \qquad\qquad (1)$$

$$117 \qquad X_t = \frac{\partial \omega}{\partial k}, \quad Y_t = \frac{\partial \omega}{\partial l} \dots \qquad\qquad (2)$$

118 Here $x$ and $y$ are the axes of the Cartesian coordinate system directed to the east and north,

119 respectively; $t$ is the time; $(k, l)$ are the components of the wave vector $\kappa$, $\omega$ is the frequency,

120 $X = X(\omega, k, l)$ and $Y = Y(\omega, k, l)$ are the ray variables in a coordinate system rotated

121 counterclockwise by an angle $\theta$.

122   Let us assume that the background flow is a stationary shear flow directed at a certain angle

123 $\theta$ fixed to the parallel. For certainty, we will consider the angle the angle $\theta > 0$ if it is counted

124 counterclockwise. To find a solution, we will proceed as follows. At the first stage, let us go over

125 to the coordinate system associated with the flow. Then in the new coordinate system rotated by

126 the angle $\theta$, the background current velocity field has only one longitudinal velocity component

127 $\vec{U} = (U, 0) = (U(y), 0)$. Further, the coordinate system is chosen so that at its origin the velocity





field is zero. Assume that $U$ is approximately linear in $y$: $U = U_y y$. Having solved the problem in
a new (rotated) coordinate system, we then make a reverse rotation by an angle $(-\theta)$, and thus we
get a solution in the original coordinate system tied to the parallel and the meridian, which is more
convenient for a clear illustration of the result.

The dispersion relation in the new coordinate system is (Gnevyshev, 2020a):

$$\omega = -\frac{\beta(k\cos\theta - l\sin\theta)}{k^2 + l^2 + F^2} + kU_y y,$$    (3)
where $\beta = \dfrac{df}{dy}$, f is the Coriolis parameter, $F^2 = \dfrac{f^2}{gH}$, g is the acceleration of gravity, $H$ is the
depth of the ocean. In the new coordinate system, there are two cyclic variables; they are the
longitudinal coordinate $x$ and time $t$. Consequently, the problem has two integrals of motion: the
longitudinal component of the momentum (in the ray approach, this is the $x$-component of the
wavenumber $\kappa$) and the wave frequency ω.

The integrated first pair of equations (1) has the form:

$$k = k_0 = const, \quad l_c = l_0 - U_y k_0 t,$$    (4)
where $(k_0, l_0)$ are the initial components of the wavenumber at $t = 0$. Note that the integrated first
pair of the equations of Hamilton gives a result that is identical to the result obtained in the
framework of the Cauchy problem (Gnevyshev et al., 2020a).

Integrating Eqs. (2), we find the coordinates of the quasi-monochromatic wave packet, at

the initial moment located at the origin of coordinates:
$$Y_\theta = \frac{\beta}{U_y}\left[\frac{\cos\theta - \sin\theta\left(\dfrac{l_0}{k_0} - U_y t\right)}{k_0^2 + F^2 + (l_0 - k_0 U_y t)^2} - \frac{\cos\theta - \sin\theta\left(\dfrac{l_0}{k_0}\right)}{k_0^2 + F^2 + l_0^2}\right]$$    (5)
$$X_\theta = \frac{\beta\cos\theta}{U_y}\left[\frac{k_0}{k_c^3}\left\{-\arctan\left(\frac{l_c}{k_c}\right) + \arctan\left(\frac{l_0}{k_c}\right)\right\}\right] - \frac{\beta\cos\theta}{U_y k_c^2}\left[\frac{F^2 U_y t + k_0 l_0}{l_c^2 + k_c^2} - \frac{k_0 l_0}{l_0^2 + k_c^2}\right] +$$

$$+ \frac{\beta\sin\theta}{U_y}\left[\frac{1}{2k_0^2}\ln\left(\frac{l_c^2 + k_c^2}{l_0^2 + k_c^2}\right) - \frac{1 - U_y t l_c k_0^{-1}}{l_c^2 + k_c^2} + \frac{1}{l_0^2 + k_c^2}\right] + U_y t Y_c$$    (6)



The subscript index $\theta$ in the solution $(X_\theta, Y_\theta)$ shows that this solution was found in a coordinate
system rotated counterclockwise by an angle $\theta$. For simplicity, the following notation is
introduced in formula (6):
$l_c = l_0 - U_y k_0 t, \quad k_c = \sqrt{k_0^2 + F^2}$ .                                    (7)
Let us turn to dimensionless variables taking into account the Rossby baroclinic radius:
$k^* = k_0 / F, \quad l^* = l_0 / F, \quad k_c^* = k_c / F, \quad l_c^* = l_c / F, \quad X^* = X_c F, \quad Y^* = Y_c F$,   and   dimensionless
time for the shear of the background flow velocity: $t^* = t |U_y|$. Omitting the asterisks, we get:
$$Y_\theta = \frac{\beta}{FU_y} \left[ \frac{\cos\theta - l_c k^{-1} \sin\theta}{k_c^2 + l_c^2} - \frac{\cos\theta - l k^{-1} \sin\theta}{k_c^2 + l^2} \right]$$                    (8)
$$X_\theta = \frac{\beta\cos\theta}{FU_y} \left[ \frac{k}{k_c^3} \left\{ -\arctan\left(\frac{l_c}{k_c}\right) + \arctan\left(\frac{l}{k_c}\right) \right\} \right] - \frac{\beta\cos\theta}{FU_y k_c^2} \left[ \frac{k l + t}{l_c^2 + k_c^2} - \frac{k l}{l^2 + k_c^2} \right] +$$
$$+ \frac{\beta\sin\theta}{FU_y} \left[ \frac{1}{2k^2} \ln\left(\frac{l_c^2 + k_c^2}{l^2 + k_c^2}\right) - \frac{1 - t l_c k^{-1}}{l_c^2 + k_c^2} + \frac{1}{l^2 + k_c^2} \right] + tY$$                    (9)

where $l_c = l - k t, \quad k_c = \sqrt{k^2 + F^2}, \quad U_y > 0$                    (10)
and $t \to -t, \quad U_y < 0$.                    (11)
This solution can be simply represented as:
$X_\theta = X_1 \cos\theta + X_2 \sin\theta, \quad Y_\theta = Y_1 \cos\theta + Y_2 \sin\theta$                    (12)
where $(X_1, Y_1)$ is the packet coordinates in the case when the flow is zonal (directed along the
parallel: $\theta = 0$), and $(X_2, Y_2)$ is the packet coordinates in the case when the flow is meridional
(directed along the meridian). It is important to note that $\theta = \frac{\pi}{2}$ for the meridional direction and
the OX$_1$ axis is directed to the north and the OX$_2$ is to the west.
$$X_1 = \frac{\beta k}{FU_y k_c^3} \left[ -\arctan\left(\frac{l_c}{k_c}\right) + \arctan\left(\frac{l}{k_c}\right) \right] - \frac{\beta}{FU_y k_c^2} \left[ \frac{k l + t}{l_c^2 + k_c^2} - \frac{k l}{l^2 + k_c^2} \right] + + t Y_1$$    (13)
$$Y_1 = \frac{\beta}{FU_y} \left[ \frac{1}{k_c^2 + l_c^2} - \frac{1}{k_c^2 + l^2} \right],$$                    (14)



$$X_2 = \frac{\beta}{FU_y}\left[\frac{1}{2k^2}\ln\left(\frac{l_c^2+k_c^2}{l^2+k_c^2}\right) - \frac{1-t\,l_c\,k^{-1}}{l_c^2+k_c^2} + \frac{1}{l^2+k_c^2}\right] + tY_2,$$
(15)

$$Y_2 = \frac{\beta}{FU_y}\left[\frac{l\,k^{-1}}{l^2+k_c^2} - \frac{l_c\,k^{-1}}{l_c^2+k_c^2}\right]\ldots$$
(16)

Then, designating the coordinates of the package in the coordinate system tied to the east and north
directions $(X, Y)$, you need to reverse the rotation of the coordinate system (counterclockwise).
Finally, we get the following expressions in a matrix form:
$$\begin{pmatrix} X \\ Y \end{pmatrix} = \begin{pmatrix} \cos\theta, & -\sin\theta \\ \sin\theta, & \cos\theta \end{pmatrix}\begin{pmatrix} X_\theta \\ Y_\theta \end{pmatrix}$$
(17)

or
$$X = X_1\cos^2\theta + (X_2 - Y_1)\cos\theta\sin\theta - Y_2\sin^2\theta$$
(18)

$$Y = Y_1\cos^2\theta + (X_1 + Y_2)\cos\theta\sin\theta + X_2\sin^2\theta$$
(19)

### 3. Numerical estimation of dimensionless parameters
We will take as the initial the following characteristic physical scales for the ocean:
$f = 10^{-4}$ s$^{-1}$, $\beta = 10^{-11}$ m$^{-1}$ s$^{-1}$, $F = 0.5 \times 10^{-5}$ m$^{-1}$. Some numerical estimates give something like this:
whereas we take for the scale of the background flow velocity $U = 5$ cm / s, and the scale of the
background flow variability 50 km, then the unit of the dimensionless time scale $U_y^{-1}$ is about 11
days. Therefore, the dimensionless time $t = 2.86 \times \pi$ is about 3 months. In this case, the
dimensionless parameter $\frac{\beta}{U_y F}$ is equal to 0.5. Whereas we take 100 km as the scale of the
background flow variability, then the unit of the dimensionless time scale $U_y^{-1}$ is approximately 22
days. Then the dimensionless time $t = 2.86 \times \pi$ is about 6 months, and the dimensionless parameter
$\frac{\beta}{U_y F}$ is equal to 1.0. These estimates make the results obtained physically justified and correct
for practical use.
### 4. Graph analysis
Qualitatively, all plots can be divided into two cases: for zonal flow (Fig. 1) and non-zonal
(Fig. 2). A common property of all graphs is that with increasing time, all rays adhere to the critical
layer. However, the number of critical layers, as well as their location, is a nontrivial function of



the angle of inclination of the background flow. Qualitatively, several main scenarios can be
distinguished.

*Zonal flow scenario.* If the flow is strictly zonal, $\theta = 0$ (Fig. 1), then one critical layer is

formed, which does not depend on the initial direction of the group velocity and is determined only
by the magnitude of the modulus of the initial wavenumber. The expression for the ordinate of the
critical layer is determined by the following (nonzero) value:
$$Y_{1c}\big|_{t\to\infty} \to -\beta\left(F U_y\left[k_c^2 + l_0^2\right]\right)^{-1} \tag{20}$$

In the case of a strictly zonal flow, all waves adhering to the critical layer move strictly to

the west: $X_{1c}\big|_{t\to\infty} \to -\infty$. It is also important to note that the movement of Rossby waves at certain
points in time is possible both to the east and in other directions. However, with increasing $t$, all
rays adhere to the critical layer, moving strictly to the west. An analysis of the tracks shows that
the dimensionless time values at which the movement begins to follow a strictly westerly direction
is approximately $t = 8$, and it gives a period of about three months for the open ocean.

In the case of a zonal flow, the initial component of the group velocity in the meridional

direction is proportional to $k_0 \times l_0$. For the zonal component of the group velocity, the sign is
determined by the following expression: $\left(k_0^2 - l_0^2 - 1\right)$. To have an idea of all possible cases, it
suffices to take the following set of four initial wavenumbers $\left(k_0, l_0\right)$. Figure 1 shows four options
for the initial direction of the group velocity; the tracks are drawn for the case $U_y > 0$. The abscissa
axis is directed to the east, the ordinate is to the north. Track 1 – the initial group velocity is directed
to the southwest. The initial components of the wavenumber are $k_0 = -1$, $l_0 = 1$. Track 2 –the initial
group velocity is directed to the southeast: $k_0 = -4\sqrt{2}/\sqrt{17}$, $l_0 = \sqrt{2}/\sqrt{17}$ or $k_0 = -1.372$, $l_0 =$
0.343. Track 3 – the initial group velocity is directed to the north-east:
$k_0 = -4\sqrt{2}/\sqrt{17}$, $l_0 = -\sqrt{2}/\sqrt{17}$ or $k_0 = = -1.372$, $l_0 = -0.343$. Track 4 – the initial group velocity
is directed to the northwest: $k_0 = -1$, $l_0 = -1$. The wavenumbers are specially selected so that the
tracks adhere to one critical layer. For all four combinations, the relation $k_0^2 + l_0^2 + 1 = 3$.


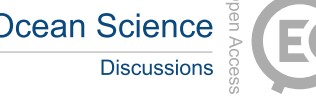

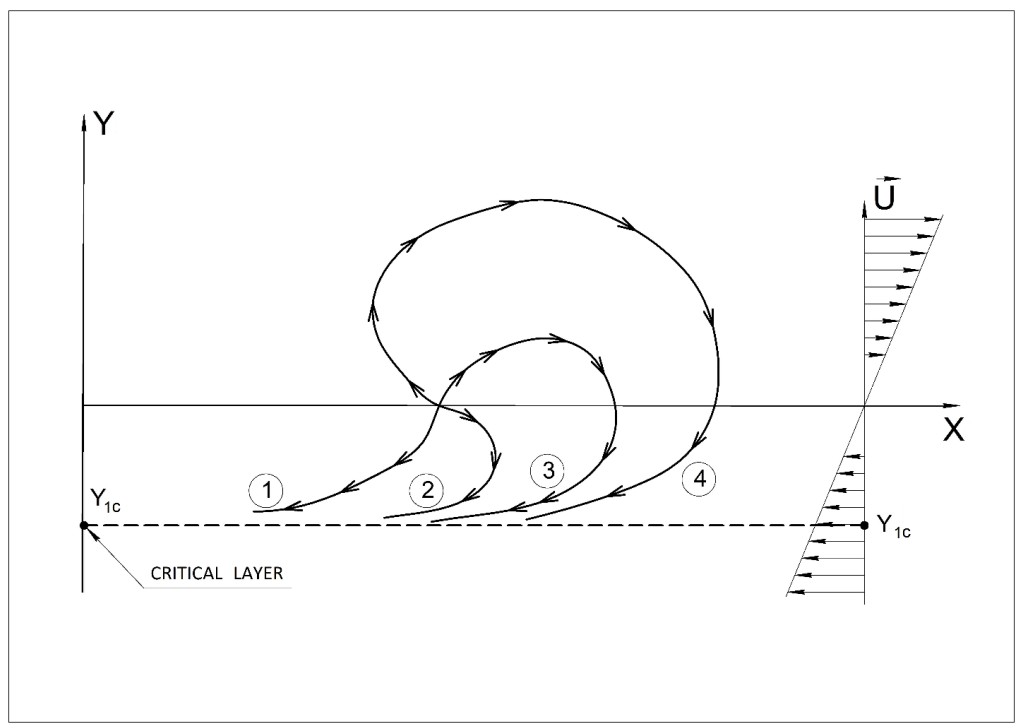


Fig. 1. The variety of tracks of Rossby waves in their interaction with the zonal flow.
Descriptions of tracks 1 - 4 are given in the text.

*Non-zonal flow scenario*. For a strictly meridional flow $\theta = \dfrac{\pi}{2}$, there are three qualitatively

different cases for the implementation of the critical layer, which can be conventionally called
"positive", "negative" and "zero". For the case of a strictly zonal flow, the critical layer is the
boundary of the transparency region. For any non-zonal flow, additional critical layers appear that
are inside the transparency region. The critical layer is "negative", for which the sign of the
intrinsic frequency adhering to the critical layer is negative. Such waves with a negative intrinsic
frequency are commonly called "waves of negative energy" (Fabrikant, Stepanyants, 1998). The
peculiarity of the non-zonal case is that Rossby waves, starting from zero value, can change the
sign of their intrinsic frequency at a certain moment in time.

The expression for the ordinate of the critical layer is determined by the following value.

$\left. Y_{2c} \right|_{t \to \infty} \to \dfrac{l_0}{k_0 U_y} \left[ \dfrac{\beta}{F \left[ k_c^2 + l_0^2 \right]} \right]$                     (21)



Recall that the coordinate system is tied to the direction of the flow velocity, so in this case,
when $\theta = \dfrac{\pi}{2}$, the $x$-axis is directed to the north and the $y$-axis to the west.
Group speed signs are defined as follows:
$C_{grx} \approx \left( -k\,l \right), \quad C_{gry} \approx \left( k^2 + 1 - l^2 \right).$
Consider the case $U_y > 0$. Provided $l\,k^{-1} > 0$, waves adhere to the negative critical layer:
$\left( Y_{2c} > 0 \right)$. Wherein $X_{2c}\big|_{t\to\infty} \to -\infty$, and the value of the group velocity along the $x$-coordinate
turns out to be negative. That is, it turns out that for adhesion to the negative critical layer, the
wave must start against the direction of the flow, but the flow will certainly turn the wave in the
direction of the flow. The wave will cross the critical layer, change the sign of its intrinsic
frequency, reflect from the higher value of the background flow velocity, and start again
approaching the critical layer, but from the opposite side. This wave behavior is called
overshooting (see Gnevyshev et al., 2020a); it also occurs in quantum mechanics.
For the initial values ($k_0 = 1$, $l_0 = 1$), the direction of the group velocity has the opposite
direction with respect to the flow, and a negative critical layer is realized. Whereas for the initial
values ($k_0 = -1$, $l_0 = 1$), the direction of the group velocity coincides with the direction of the flow,
and the negative critical layer is not realized. Reflection occurs, and the wave goes to the positive
critical layer.
Provided $l_0\,k_0^{-1} < 0$, waves adhere to the positive critical layer, $\left( Y_{2c} < 0 \right)$. The situation is
qualitatively similar to the purely zonal case. In this case, the critical layers have not only
components of different signs and magnitude, but also tend to $\pm\infty$ by the $x$-coordinate,
$\left( X_{2c} \to -\infty \right)$.
From the analysis of these ratios, it can be seen that an additional second critical layer,
which appears due to the non-zoning of the flow. is realized only for waves that initially fall strictly
against the current. Whereas waves that fall in the direction of the flow have a trivial reflection
from the negative critical layer. Let us also note the existence of a third scenario. At $l_0 = 0$, the
wave starts strictly perpendicular to the background current, while the critical layer $\left( Y_{2c} = 0 \right)$ is
zero.
Let us analyze the intermediate flow direction. The asymptotics for the ordinate of the
critical layer in the general case has the form:



$$Y_\theta\big|_{t\to\infty} \to \frac{l_0 \sin\theta - k_0 \cos\theta}{k_0 U_y}\left[\frac{\beta}{F\left(k_0^2 + l_0^2\right)}\right]\ldots$$   (22)
The longitudinal component of the group velocity is proportional to
$\left(k_0^2 - l_0^2 - 1\right)\cos\theta - 2k_0 l_0 \sin\theta$ .
The transverse component of the group velocity is proportional to
$2k_0 l_0 \cos\theta - \left(l_0^2 - k_0^2 - 1\right)\sin\theta$ .
It follows from expression (22) that when even weak non-zonality appears, there is not one,
as in the case of a purely zonal flow, but three critical layers since the value $\left(l_0 \sin\theta - k_0 \cos\theta\right)$
can be positive or negative values or zero. For zonal flow, regardless of the parameters of the
wavenumber of the incident wave, any wavenumbers can be considered, however, the critical layer
is always at negative velocities. For a non-zonal flow at different wavenumbers, that is, at different
angles of incidence on the flow, there will be three such critical layers: one at a negative velocity
value, one at a positive velocity value, and one with zero velocity. If we fix the wavenumber, then
there is always one critical layer. For a zonal flow, this layer will correspond to a negative velocity
value. For non-zonal flow, there are possible options: the critical layer will be located either at a
positive velocity value or at a negative one or with zero velocity. In other words, some
wavenumbers will stick to the positive, and others to the negative values of the background
velocity. When we say "one critical layer", we do not mean a fixed value of the velocity, but only
its sign.
The first critical layer that is implemented for western propagation is the classic well-
known and well-studied critical layer for Hermitian operators. The second critical layer is realized
for waves moving eastward. This critical layer does not have symmetries due to the non-Hermitian
nature of the non-zonal linear operator of Rossby waves and introduces such a phenomenon as
overshooting into the kinematics of Rossby waves. The third critical is zero and is inherent only
in strictly non-zonal flows. In this scenario, the waves return to the initial level from which they
started.
For simplicity of numerical values, we take the angle $\theta = \dfrac{\pi}{4}$ . Then we have the following
typical sets of wave tracks: track 1 – ($k_0$ = - 0.5, $l_0$ = 1); track 2 – ($k_0$ = -1, $l_0$ = 1); track 3 – ($k_0$ = -
2, $l_0$ = -0.5); track 4 – ($k_0$ = -1, $l_0$ = -2); track 5 – ($k_0$ = -1, $l_0$ = -1). Such a variety of possible
scenarios is inherent precisely in Rossby waves and is associated with the absence of symmetries


in the problem, which are a consequence of the non-Hermitian nature of the linear operator of
Rossby waves for arbitrary shear flows.

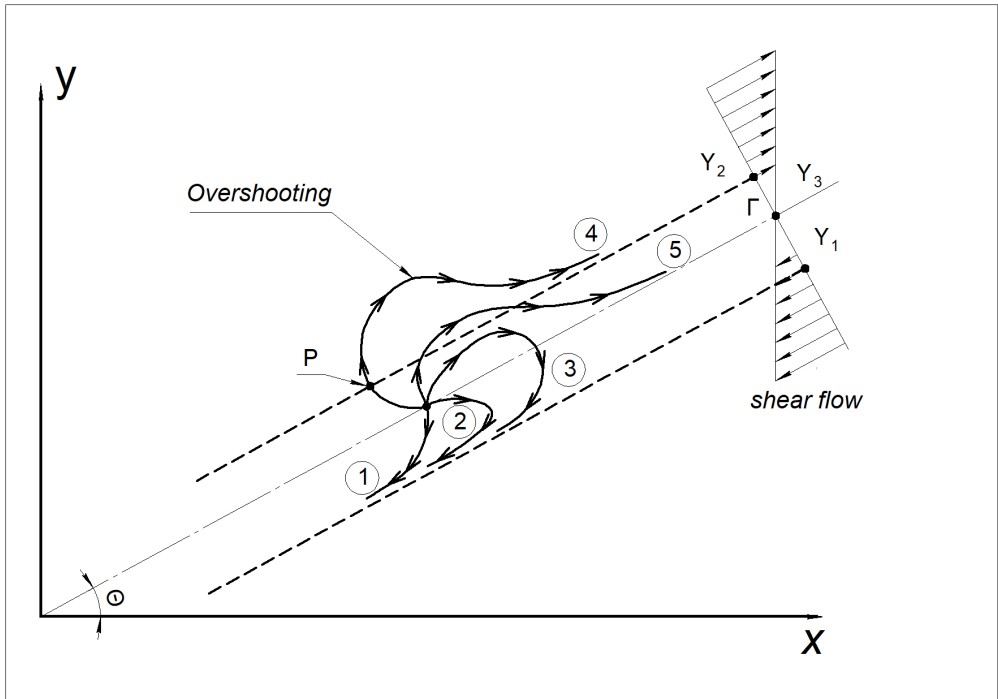


**Fig. 2.** The variety of Rossby wave tracks in their interaction with the non-zonal current. The

description of tracks 1 - 4 is given in the text.

**Discussion and Conclusions**

The ray equations of Hamilton are a kind of approximate method for analyzing the

kinematics of waves. Therefore, a question arises: what are the limits of applicability of these
equations?

To answer this question, we will proceed from the statement that, from a mathematical

point of view, the solution of the Cauchy problem is more correct than the ray equations of
Hamilton. The solution of the Cauchy Problem for Rossby waves on a shear plane-parallel flow,
in a coordinate system associated with the flow and directed at a certain angle $\theta$ to parallel, has
the form (Gnevyshev et al., 2020a, b):



$$\Psi(x,y,t) = \int\limits_{-\infty}^{+\infty} \int\limits_{-\infty}^{+\infty} G(k,l) \frac{\left(k_z^2 + l^2\right)}{\left(k_z^2 + l_t^2\right)} \times \exp\left(i\Upsilon(x,y,k,t)\right) dk\, dl\, ,$$
$$\Upsilon(x,y,k,l,t) \equiv \frac{\beta\cos\theta}{U_y k_z} \left\{ -\arctan\left(\frac{l_t}{k_z}\right) + \arctan\left(\frac{l}{k_z}\right) \right\} + \frac{\beta\sin\theta}{2U_y k} \ln\left(\frac{k_z^2 + l_t^2}{k_z^2 + l^2}\right) +$$
$$+ \left[ k\left(x - U_y yt\right) + ly \right]$$ (23)

where the following designations are introduced: $l_t = l - U_y kt$, $k_z = \sqrt{k^2 + F^2}$. We construct the
phase for the solution in the form of the ray equations as follows:
$$\Theta(x,y,k,l,t) = -\int \omega\, dt$$ (24)
Let us substitute in (24) the expression for the frequency (3) and the first pair of integrated
equations (4). In this case, using the free term in the form of an arbitrary function of the
wavenumbers, we normalize the phase as follows: $\Theta(y,k,l,t)\big|_{t=0} = 0$. Integrating (24) with the
chosen normalization conditions, we obtain:
$$\Theta(y,k,l,t) = -\int \left\{ \frac{-\beta\left(\kappa_0\cos\theta - l_c\sin\theta\right)}{\kappa_0^2 + l_c^2 + F^2} + \kappa_0 U_y y \right\} dt =$$
$$= \frac{\beta\cos\theta}{U_y \kappa_z} \left\{ -\arctan\left(\frac{l_c}{\kappa_c}\right) + \arctan\left(\frac{l_0}{\kappa_c}\right) \right\} + \frac{\beta\sin\theta}{2U_y \kappa_0} \ln\left(\frac{\kappa_c^2 + l_c^2}{\kappa_c^2 + l^2}\right) - \kappa_0 U_y yt$$ (25)

Comparing the obtained expression (25) for the normalized phase of the WKB-solution with the
expression for the phase of solution (23) of the Cauchy problem, we find the following relation:
$$\Theta(y,k,l,t) + kx + ly = \Upsilon(x,y,k,l,t) .$$
Thus, the phases of the solutions coincide. On the other hand, if we assume that the scale of
changes in the main flow is much larger than the characteristic scale of the solution for
perturbations, then a small parameter ε will appear in the problem (Gnevyshev et al., 2019, 2021),
which formally, after reduction to dimensionless form, is expressed by replacing the derivative the
main flow velocity $U_y$ by $\varepsilon \times U_y$. Passing in the expression for the phase of solution (23) to the limit
in $U_y$, as in a small parameter, and keeping the zero and first terms of the expansion, we obtain the
following relation:
$$\Upsilon(x,y,k,l,t)_{(U_y t \to 0)} \to \left( \frac{-\beta\left(\kappa\cos\theta - l\sin\theta\right)}{\kappa^2 + l^2 + F^2} + \kappa U \right) t + \kappa x + ly = \omega t + \kappa x + ly ,$$





where $\omega = \dfrac{-\beta\left(\kappa\cos\theta - l\sin\theta\right)}{\kappa^2 + l^2 + F^2} + \kappa U$ .
On the other hand, from (23) it is easy to obtain the following relation:
$\lim \Upsilon(x, y, k, l, t)_{(U_y t \to \infty)} \neq \omega\, t + \kappa\, x + l\, y$

Summing up, let us emphasize the first original result obtained in this work. Solutions (5)

and (6) obtained in the framework of the Cauchy problem are exact solutions of ray equations (1)
and (2). Consequently, not only do the limiting values obtained within the framework of the WKB-
solution and the Cauchy problem in the first approximation coincide, but also the solutions
themselves. In other words, the integral of the solution phase, obtained in the first order of the
WKB approximation and normalized to zero at the initial moment of time, coincides with the phase
of the basic solution of the Cauchy problem. In this case, the expansion of the phase of the solution
of the Cauchy problem in terms of the small WKB-parameter in the first approximation gives the
dispersion relation obtained in the first order of the WKB-solution. For large time intervals, the
phase of the solution to the Cauchy problem does not reach the WKB-solution mode. Hence, from
the point of view of the Cauchy problem, the WKB-solution cannot work up to any infinitely large
times with a finite shear of the background flow velocity profile.

Otherwise, it can be explained as follows. The time $t$ and the shear of the background

current velocity $U_y$ are included in the solution in the form of the product $t \times U_y$. Consequently,
whatever the small parameter $U_y$, there will come a time t such that the product $t \times U_y$ will be
greater than one, and the series expansion of the solution phase will no longer be justified.

Thus, the application of the Hamiltonian formalism in a linear problem helps to build a

bridge between seemingly different solutions obtained in the WKB-approximation and the
framework of the Cauchy problem. In this case, the first pair of ray equations (1) is nothing but
the condition of equality of the cross derivatives of the solution phase. The second pair of ray
equations (2) is the equation for a stationary point. The mathematical reason for this behavior is
that in the presence of non-zoning in the solution phase, a logarithm of the form appears
$\ln\left(1 + U_y^2 t^2\right)$. The Taylor series of the logarithm at zero has a radius of convergence equal to one.
Consequently, no matter how small the value of the shear in the profile of the background flux $U_y$
is, there will come a time at which the argument of the logarithm will exceed one and the
asymptotic expansion will stop working.

In this paper, using the example of Rossby waves on non-zonal shear flows, explicit

analytical integration of the ray equations of Hamilton is performed for the first time. Previously,





no one paid attention to this possibility. It turned out that the obtained explicit analytical solution
of ray equations of Hamilton is expressed in simple elementary functions, which turned out to be
quite unexpected. The constructed typical kinematic tracks of Rossby waves on non-zonal shear
currents show the relevance of such a phenomenon as the critical layers of Rossby waves.
In its simplicity and ease, this method surpasses the solution in terms of the Cauchy
problem using convective coordinates, and from an analytical point of view, it is identical to the
asymptotics of the two-dimensional integral of the Cauchy problem that we obtained earlier
(Gnevyshev et al., 2020a).
An analytical comparison of the obtained solution with the solution of the Cauchy problem
for Rossby waves is made. For small time intervals, the solutions of the ray equations strictly
coincide with the asymptotics of the integral obtained in the framework of the Cauchy problem.
The non-zonality of the flow leads to the appearance of a logarithm in the solution phase, which
greatly complicates the convergence of the results obtained. At large time intervals, the non-
zonality of the flow leads to a logarithmic spreading of the solution, which requires additional
analysis within the framework of the convolution of the obtained solutions over the spectrum of
wavenumbers.
The obtained analytical expressions were used to construct the kinematic tracks of Rossby
waves on shear flows. The solutions are anisotropic and, in the general case, do not have classical
north-south symmetries.
It is shown that in the non-zonal case, a second critical layer is added to the classical critical
layer of Rossby waves for the strictly zonal case, which is directly related to such concepts as
negative energy waves and overshooting.
**Acknowledgements**
The publication was funded by the Russian Science Foundation, project No 22-27-00004.
**Author Contributions**
VG presented the idea, made theoretical analysis, wrote the paper draft. TB plotted figures,
wrote and edited the manuscript. All authors have read and agreed to the published version of the
manuscript.
**Funding**
The publication was funded by the Russian Science Foundation, project No 22-27-00004.



**Compliance with Ethical Standards**

**Conflict of interest**

The authors declare that they have no conflicts of interest.

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
