# Peer review of "Analytical solution of the ray equations of Hamilton for Rossby waves on stationary shear flows"

_Ocean Science, 2022_

## Referee Comment (RC1)

Referee's report on OS-2022-5 "Analytical solution of the ray equations of Hamilton for Rossby waves on stationary shear flows" by V. G. Gnevyshev and T. V. Belonenko

The manuscript describes numerical solutions of the ray equation associated with Rossby waves embedded in a mean shear flow. As expected, the solutions of the autonomous, nonlinear, fourth order system of ordinary differential equation, depend sensitively on the initial conditions, which is manifested in different types of trajectories. Despite the simplicity of the mathematical system I cannot recommend publication of this manuscript in Ocean Science for the general reasons listed below.

General

1. The main issue I have with this work is that it teaches us nothing new about "Ocean Science". The authors make no attempt to relate the solutions they compute to an oceanographic phenomenon. I'm not sure whether this work warrants publication as a contribution in applied mathematics or fluid mechanics but it is definitely not a contribution in oceanography. The authors do not attempt to contribute anything to our understanding of phenomena observed in physical oceanography.

2. From a mathematical standpoint, it is rare for nonlinear, $4^{th}$ order, systems to poses analytic solutions. However, for the most part, numerical solutions are both easy to compute and very accurate (unless there is a singularity in the problem which is not the case here). The ray equation is no more than the calculation of trajectories in space when the (group) velocity is spatially variable (including the spatial variability of the wavenumber). It is unclear what has been gained from the few trajectories calculated in this manuscript. The main message of this paper is too thin mathematically. As highlighted by the authors, the attraction of all trajectories to the critical point (e.g. latitude) is well known and was highlighted in previous papers by the same authors (Gnevyshev et al., 2020a; 2020b). Solutions of such basic mathematical systems are publishable only when they constitute significant contributions to our understanding of oceanographic processes.

3. The description of the underlying assumptions and the methodology employed is very poor. The extremely weak pedagogical presentation typifies both the English style and the mathematical analysis. If the authors wish to publish this work in a different journal they should tend much more seriously to both of these aspects of communication with their readers. A few examples are listed below as an aid to the authors if they intend to submit the manuscript to another journal.

Particular points:

Below I list examples of the poor presentation of the (fairly simple) material. The intent in this list is to give the authors (whose mother tongue is probably Russian) a sense of what is expected from a well presented scientific contribution.

1. Mathematical Analysis

a. The x and y coordinates in (1) are usually identical with the X and Y coordinates in (2). The ray equation applies the DEL operator on \omega to calculate the temporal changes in the wavenumbers using the same coordinates in which the advection is calculated. If this case is special please elaborate.

b. The non-dimensionalization should be the first item following equation (3). Carrying it out after the development of the explicit solutions in (5) and (6) in dimensional variables is very confusing, leading to the confusion between dimensional and non-dimensional variables.

c. The non-dimensional expressions cannot contain dimensional variables. The non-dimensional equations (8) and (9) contain the dimensional parameters \beta, $F$ and $U_y$ though the system was made non-dimensional beforehand.

d. In equation (10) the dimensional $F$ cannot be added to the non-dimensional $k$ (which was non-dimensionalized after (7)).

e. The message in (11) is unclear. What do you mean?

f. Your value of $F$ implies a radius of deformation of 200 km. It does not make sense to use scales of the domain that are 50 km or 100 km only.

g. On the small meridional scales you chose, Rossby waves cannot be observed (or expected to prevail).

h. In Fig. 2 there are 5 trajectories not 4 as you describe in the caption!

i. Are the initial conditions you use exhaustive? Can one find additional types of trajectories by starting (say, in Fig. 1) with a northward initial velocity? One will definitely encounter a different trajectory starting with a zonal initial velocity.

j. What happens if the trajectory emanates from X>0 or X<0?

2. English style

The English style of the manuscript is below the acceptable standard as it requires the readers to invest heavily in reading it. I list only a few examples:

a. Lines 15-16: *On the example of Rossby waves on a shear flow, the ray equations of Hamilton are analytically integrated* – This is not an English sentence
b. Lines 35-36: *One of the central moments in the interaction of Rossby waves and large-scale flows are critical layers* – No time (moment as in mnute) or mechanics (e.g. moment of inertia) intended here
c. Line 36: *The classical critical layer is not formally attainable for waves* – Attainable (as in Gain, Achieve or Obtain) does not belong here.
d. Line 182: *Whereas* – Whereas (as in While, Although or Since) does not belong here.
e. Lines 220-225: "*Adhering*", "*Transparency*", "*negative*" and "*for which*" – out of place and if have to be used should be explained.
f. Lines 250-254: a wave never "*falls*" on a current (or on anything else).
g. Line 287: *inherent precisely* – What do you mean?
h. Lines 298-326 – The long mathematical expansion does not belong in the Discussion and Conclusion section.